# Geographic spillover of antimicrobial resistance from mass distribution of azithromycin

Ariktha Srivathsan [1,2], Ahmed M. Arzika[3], Ramatou Maliki[3], Amza Abdou[4], Marc Lipsitch [5,6], Seth Blumberg [1,7], Kieran S. O'Brien[1,2,8,9], Travis C. Porco[1,2,8], Armin Hinterwirth [1], Thuy Doan [1,8], Jeremy D. Keenan [1,8], Thomas M. Lietman [1,2,8,9] & Benjamin F. Arnold [1,8,9] ✉

Large-scale, placebo-controlled, cluster-randomized trials in high-mortality settings in sub-Saharan Africa demonstrated a 14–18% reduction in childhood mortality following twice-annual mass drug administration (MDA) of azithromycin among children aged 1–59 months. Azithromycin MDA also selected for antimicrobial resistance (AMR), particularly macrolide resistance. It is unknown whether the AMR from azithromycin MDA could spill over to neighboring untreated populations. If present, such geographic spillover effects could lead trials to underestimate AMR risks. We assess between-village geographic spillover effects of genotypic macrolide resistance using metagenomic deep sequencing in rectal swabs collected from 300 children in 30 monitoring villages in Niger after two years of MDA in 594 surrounding villages. Conditional permutation tests assess associations between proximal azithromycin treatment intensity and resistance gene abundance. We find no evidence of geographic spillover of macrolide resistance in untreated villages, as the genetic load of AMR remains at baseline levels in placebo-treated villages regardless of surrounding azithromycin treatment intensity (Spearman $\rho = -0.05$, $P = 0.83$). Sensitivity analyses confirm robustness across metrics, and no spillover effects are detected for other antibiotic classes. Azithromycin MDA-induced macrolide resistance appears localized to treated villages, mitigating some concerns about geographic spillover of AMR to nearby untreated villages at 24 months.

In 2018 the MORDOR (Macrolides Oraux pour Réduire les Décès avec un Oeil sur la Résistance) trial demonstrated a 14% reduction in all-cause mortality among children 1–59 months following twice-annual mass drug administration (MDA) with azithromycin[1]. This evidence resulted in the World Health Organization (WHO) issuing conditional recommendations that azithromycin MDA be considered for children in high-mortality settings, but the WHO limited the recommendation to children aged 1–11 months to help mitigate the potential for selection of antimicrobial resistance (AMR) from MDA[2]. Subsequent trials in Burkina Faso[3] and Niger[4] demonstrated mortality reduction when twice-annual MDA is delivered to all children 1–59 months, but not if delivery was limited to children 1–11 months[4] or if delivered to individual children postnatally[5] or at well-child visits[6]. Based on this evidence, the government of Niger is now considering national-scale, biannual azithromycin MDA to reduce child mortality[7].

Repeated azithromycin MDA to children aged 1–59 months selects for AMR, particularly to macrolides. Despite the evidence of increased macrolide resistance within treated populations[8–10], it remains unclear whether selection of macrolide resistance could extend to larger spatial scales, potentially transmitting AMR genes to neighboring, untreated populations through person-to-person transmission of resistant bacteria or, potentially, through environmentally mediated transmission. Such spillover or indirect effects[11] of AMR could potentially magnify risks beyond what is currently known[12].

Standard analyses of randomized controlled trials assume independence between randomized units (participants or clusters)[11]. Independence implies that no interference occurs between treated and untreated groups such that the outcome of a randomized unit depends solely on its own treatment, unaffected by the treatment administered to other units[13]. In azithromycin MDA trials, if between-cluster spillover effects are present, the independence assumption does not hold, and the trials could underestimate potential harms: AMR selected in treated villages could spread to untreated villages, making the two groups look more similar than they would in the absence of spillover. Infectious disease intervention trials have quantified such spillover effects across various contexts[11,14–16]. Evidence from HIV prevention[17] and antibiotic resistance to malaria control[18] suggests that even limited interaction between populations can substantially impact intervention outcomes and resistance patterns, with effects observed up to several kilometers away from intervention areas. Notably, phylogenetic analyses of HIV transmission patterns demonstrate that population mobility and geographic proximity are fundamental drivers of spread between neighboring communities[19]. Modeling studies suggest that antimicrobial resistance can spread between populations through person-to-person transmission of resistant bacteria, with even modest interactions between populations[12]. Concerns have been raised about the spillover of AMR from treated to untreated groups within communities receiving azithromycin MDA[20], with some evidence suggesting resistance may persist and potentially spread to untreated populations[21]. Understanding whether azithromycin MDA can lead to geographic spillover of AMR is crucial for assessing the broader public health implications and potential risks as countries consider scaling up azithromycin MDA to millions of children.

We conducted a pre-specified secondary analysis of the MORDOR trial in Niger (NCT02047981; primary outcomes published in ref. [1]), focusing on the potential between-village spillover effects of azithromycin MDA on AMR to macrolides and other antibiotic classes. We hypothesized that if geographic spillover effects were present, placebo-treated communities surrounded by high levels of azithromycin MDA would have a higher genetic load of AMR compared with placebo-treated communities without nearby azithromycin MDA. To test this hypothesis, we leveraged the randomized geographic variation in the distribution of azithromycin doses across 594 villages included in the trial.

## Results

### MORDOR trial: study design and study population

MORDOR was a cluster-randomized, placebo-controlled trial conducted in Niger's Dosso region (Fig. 1A) to study the effect of twice-annual azithromycin MDA on childhood mortality. The mortality monitoring trial enrolled 594 villages, with 303 assigned to receive azithromycin and 291 assigned to receive a matching placebo twice annually over a 24-month period. A random sample of 30 additional villages from the original sampling frame were enrolled concurrently to monitor the impact of azithromycin MDA on AMR. Villages in the AMR monitoring sample were also randomized to receive azithromycin ($N = 15$) or placebo ($N = 15$) (Fig. 1B). All children aged 1–59 months residing in trial villages were eligible to receive a dose of azithromycin or placebo twice annually, based on village treatment assignment.

We investigated the potential for macrolide resistance to spread beyond treated populations in the context of MDA programs. Mass distribution of azithromycin could exert selective pressure not only within treated populations but also in nearby untreated areas, influenced by patterns of human mobility, environmental factors, and microbial transmission. In the presence of between-village spillover effects, we expect the resistance in a given village to also depend on the number of azithromycin doses distributed in neighboring villages. We hypothesized the effect would be more pronounced in untreated monitoring villages, as these areas would not experience the direct selective pressure of treatment but would still be affected by spillover. We additionally hypothesized that spillover effects could influence AMR in treated monitoring villages by augmenting resistance selection pressure beyond direct treatment. To systematically examine the relationship between spatial variation in azithromycin treatment and resistance selection, we assessed whether AMR increased with higher azithromycin distribution in neighboring villages.

### Geographic variation in azithromycin treatment intensity

We anticipated spatial heterogeneity in the number of antibiotic doses distributed across Dosso due to underlying variation in settlement patterns and the randomized assignment of azithromycin treatment to villages. We estimated an azithromycin geographic treatment intensity variable to represent the potential for geographic spillover of AMR resulting from azithromycin MDA in nearby villages. We quantified geographic treatment intensity at any point in the study region as the cumulative number of doses of azithromycin distributed in mortality-monitoring villages over 24 months (Fig. 1C), weighted by the inverse of the distance to that point. We used an inverse distance weighting approach with a linear decay function to balance the influence of nearby and distant treated villages, aligning with the expected antimicrobial resistance spread. There was substantial variation in the geographic treatment intensity variable over the study region and across the 30 monitoring villages (Fig. 1D). AMR was measured at baseline (prior to treatment) and again at 24 months, following four rounds of azithromycin distribution, in 30 monitoring villages. Metagenomic deep sequencing was performed on rectal swab samples from 10 children per village, and AMR was quantified as village-level average normalized read counts of macrolide–lincosamide–streptogramin (MLS) resistance determinants at AMR-monitoring villages.

### Spillover effects of azithromycin MDA to nearby villages

MLS resistance determinants remained low and at baseline levels in placebo-treated AMR monitoring villages at 24 months, regardless of azithromycin geographic treatment intensity (Fig. 2A, B). We quantified the association between MLS resistance and azithromycin geographic treatment intensity using Spearman rank-order correlation. To assess statistical significance, we applied permutation tests in which we permuted the treatment assignments of surrounding mortality-monitoring villages, using a conditional permutation approach[22,23]. Azithromycin treatment intensity was uncorrelated with macrolide resistance at both timepoints ($P = 0.99$ at baseline, $P = 0.83$ at 24 months) in placebo-treated villages. These results are consistent with no geographic spillover of azithromycin-induced resistance in untreated monitoring villages.

As reported previously[8], azithromycin-treated AMR-monitoring villages experienced a sharp increase in MLS resistance determinants from baseline to 24 months (Supplementary Fig. 9A), attributable to the direct effects of azithromycin MDA. A modest correlation between geographic treatment intensity and MLS resistance in azithromycin-treated villages (Fig. 2A, B) was observed at baseline ($\rho = 0.30$) and remained similar at 24 months ($\rho = 0.32$), though neither was significant under a conditional permutation test ($P = 0.21$ at baseline, $P = 0.95$ at 24 months). It is therefore likely that the observed, modest correlation reflects pre-existing spatial patterns rather than an

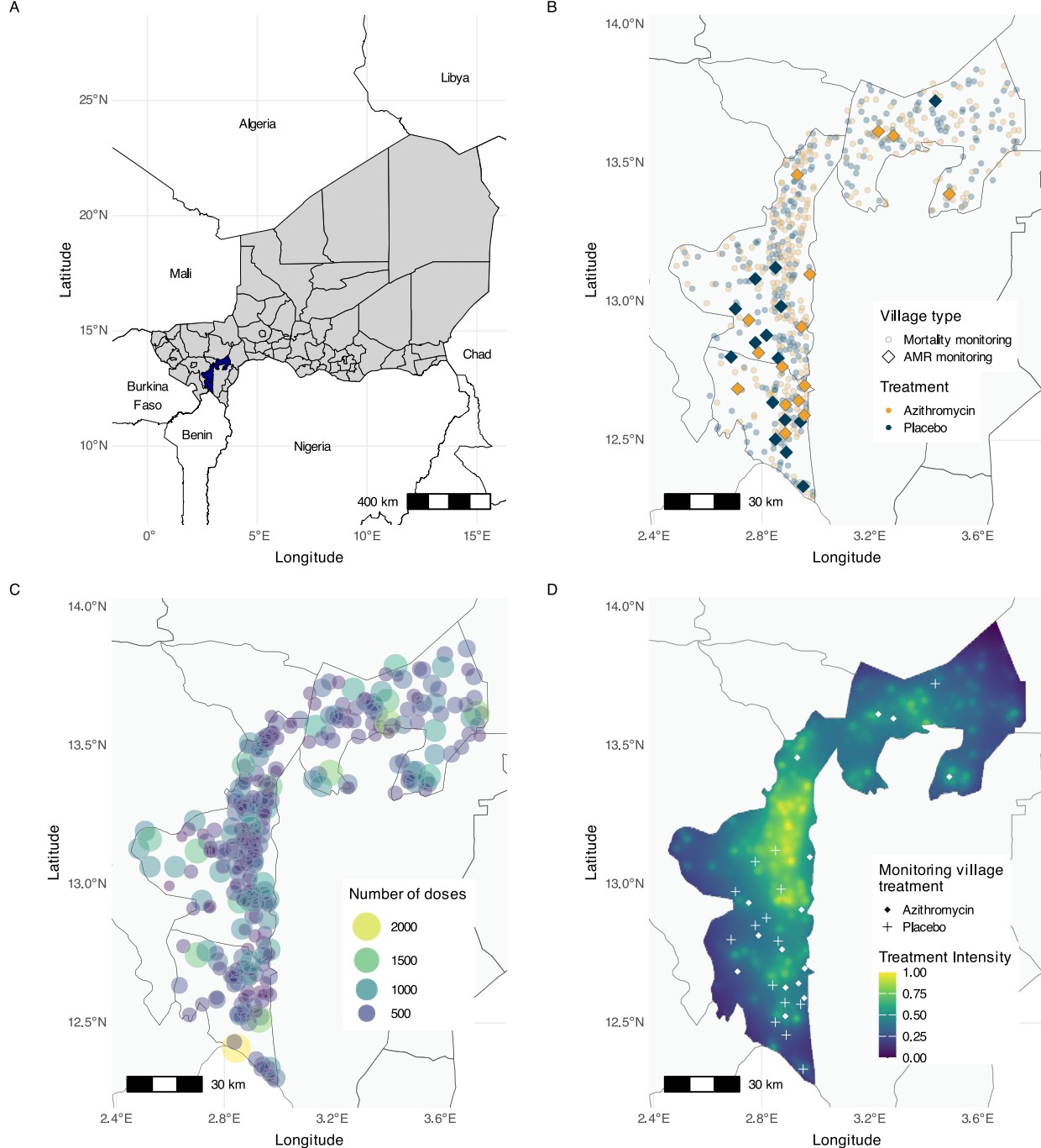

**Fig. 1 | Study area and treatment distribution in the MORDOR trial in Dosso, Niger. A** The study area includes the Boboye, Loga, and Falmey departments (highlighted in blue) in the Dosso region of Niger. The country of Niger is shown in grey, with black lines indicating department boundaries within the grey-shaded area. **B** Each point in the study area represents a village enrolled in the trials, with azithromycin-treated villages being represented in yellow and placebo-treated villages in blue. 594 villages (lighter circles) were randomized to receive mass drug administration (MDA) of azithromycin (*N* = 303) or placebo (*N* = 291) as part of the MORDOR trial to monitor childhood mortality. 30 other villages (darker diamonds) were separately randomized to receive azithromycin (*N* = 15) or placebo (*N* = 15) to monitor anti-microbial resistance. Grey lines depict department boundaries. **C** Each circle represents a mortality monitoring village treated with azithromycin (yellow circles in **B**). The circle area and color represent the number of doses distributed in the village, with smaller purple circles representing fewer doses and larger yellow circles representing higher number of doses. **D** The geographic treatment intensity layer was estimated as the inverse distance weighted sum of azithromycin doses distributed in nearby mortality monitoring villages, based on data presented in (**C**). Points mark the location of the 30 morbidity monitoring villages, with shapes indicating the treatment assignment. Administrative boundaries used in maps were sourced from the United Nations Office for the Coordination of Humanitarian Affairs (OCHA) *Common Operational Datasets – Administrative Boundaries*, accessed via the Humanitarian Data Exchange on 9 April 2024, and are licensed under CC BY-IGO 3.0 (https://data.humdata.org/dataset/cod-ab-ner). Figure created using script https://osf.io/jrvsw. Source data are provided as a Source Data file.

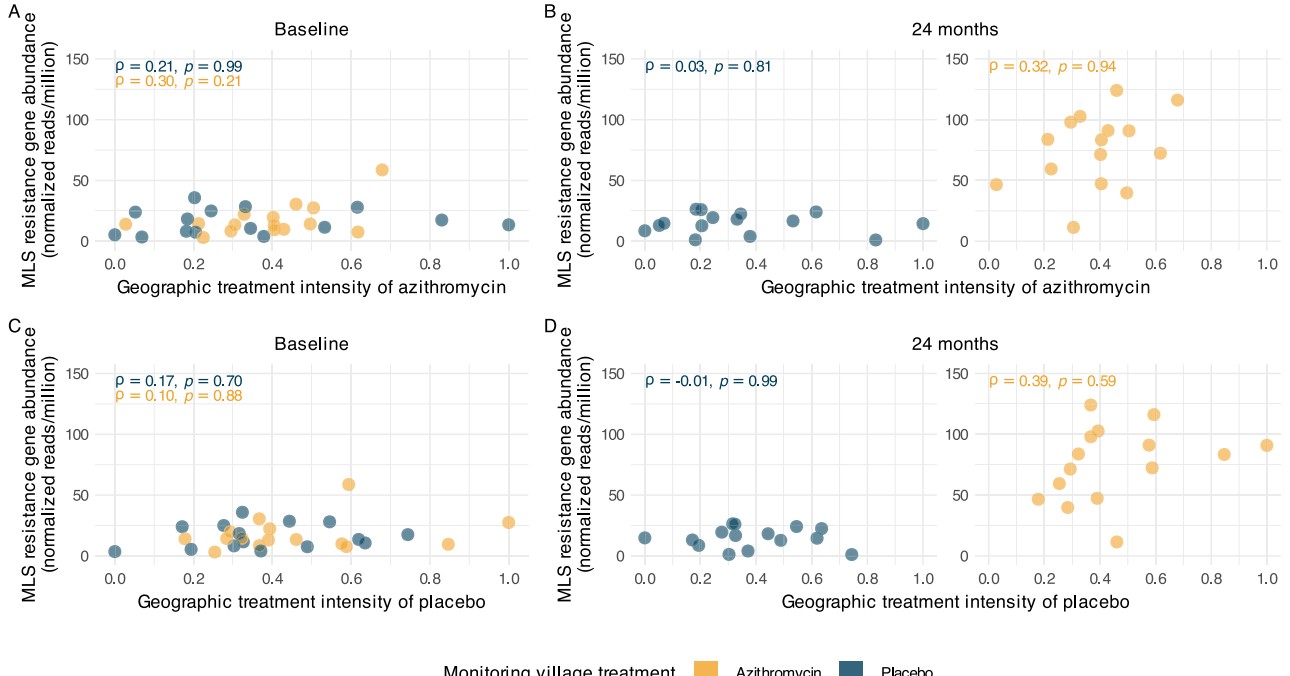

**Fig. 2 | Spillover effects of mass drug administration (MDA) with azithromycin on genotypic resistance to Macrolides–lincosamides–streptogramins (MLS).** **A** Relationship between the azithromycin treatment intensity and the village-level normalized reads of MLS resistance determinants (in reads per million) at baseline, estimated as Spearman rank-order correlations (ρ). Points represent anti-microbial resistance (AMR) monitoring villages, with azithromycin-treated villages in yellow and placebo-treated villages in blue. **B** Relationship between the azithromycin treatment intensity and the village-level normalized reads of MLS resistance determinants (in reads per million) at 24 months with placebo-treated villages in the left panel and azithromycin-treated villages on the right. Left panel: In placebo-treated villages, MLS AMR levels remained low across all values of geographic treatment intensity, indicating no evidence of spillover of resistance following MDA. Right panel: Higher MLS AMR levels were observed in the azithromycin-treated villages at 24 months as compared to the baseline. Spearman correlations in azithromycin-treated villages remained similar at baseline (ρ = 0.30) and at 24 months (ρ = 0.32), indicating a stable relationship across time points. **C** Relationship between the placebo treatment intensity (negative control exposure) and the village-level normalized reads of MLS resistance determinants at baseline. **D** Relationship between the placebo treatment intensity (negative control exposure) and the village-level normalized reads of MLS resistance determinants at 24 months with placebo-treated villages in the left panel and azithromycin-treated villages on the right. Left panel: In placebo-treated villages, MLS AMR levels remained low across all values of placebo treatment intensity, with no significant correlation being observed. Right panel: Similar to (**B**), Spearman correlations in azithromycin-treated villages indicate a monotonic trend of MLS AMR with placebo treatment intensity. The effect of placebo treatment intensity being similar to azithromycin treatment intensity indicates that the observed modest and non-significant correlation in (**B**) is likely not attributable to azithromycin distributions. All *p*-values are two-sided and derived from a permutation test. Figure created using script https://osf.io/4puya. Source data are provided as a Source Data file.

emergent spillover effect of azithromycin MDA. Additionally, when examining the change in MLS resistance from baseline to 24 months, no significant association was observed with geographic treatment intensity in either placebo (ρ = −0.09, *P* = 0.95) or azithromycin-treated villages (ρ = 0.12, *P* = 0.64, Supplementary Fig. 9B), consistent with the absence of evidence for geographic spillover effects.

## Negative control exposure analysis

As an additional robustness check, we estimated the geographic treatment intensity of placebo doses as a negative control exposure (Fig. 2C, D). Using the same methods used to estimate azithromycin geographic treatment intensity, but with the cumulative number of placebo doses distributed instead of the cumulative azithromycin doses, the negative control exposure analysis was designed to detect spurious associations arising from unmeasured confounding, or other non-causal sources of variation in AMR that are unrelated to azithromycin MDA[24,25]. As with the geographic treatment intensity of azithromycin, we observed a modest positive correlation between MLS resistance determinants and geographic treatment intensity of placebo in azithromycin-treated monitoring villages at 24 months (ρ = 0.39); however, this association was also not statistically significant under a conditional permutation test (*P* = 0.61, Fig. 2D). The change in MLS resistance from baseline to 24 months similarly showed no significant association with placebo treatment intensity in either

treatment arm (ρ = −0.03, *P* = 0.57 in placebo-treated villages; ρ = 0.3, *P* = 0.59 in azithromycin-treated villages; Supplementary Fig. 9C). Since placebo treatment is not expected to have any biological impact on MLS resistance determinants, the similarity of the negative control analysis and the primary analysis suggests that the observed non-significant correlations are unlikely to result from the spillover effects of azithromycin MDA. We estimated a partial Spearman rank correlation between geographic treatment intensity of azithromycin and AMR, conditioning on the negative control exposure (partial ρ = 0.1, *P* = 0.7) in azithromycin-treated AMR-monitoring villages at 24 months. The unadjusted correlation was already weak and non-significant (ρ = 0.32, *P* = 0.99) and further decreased in magnitude after conditioning on the negative control, reinforcing that geographic variation in azithromycin MDA intensity is unlikely to be a driver of AMR patterns.

## Spillover effects on phenotypic resistance

To complement the metagenomic analyses of resistance determinants in rectal swabs, we assessed phenotypic macrolide resistance in *Streptococcus pneumoniae* isolated from nasopharyngeal swabs collected at 24 months. We evaluated the relationship between geographic treatment intensity and the proportion of *S. pneumoniae* isolates resistant to erythromycin using Spearman rank-order correlation, with statistical significance assessed through permutation tests.

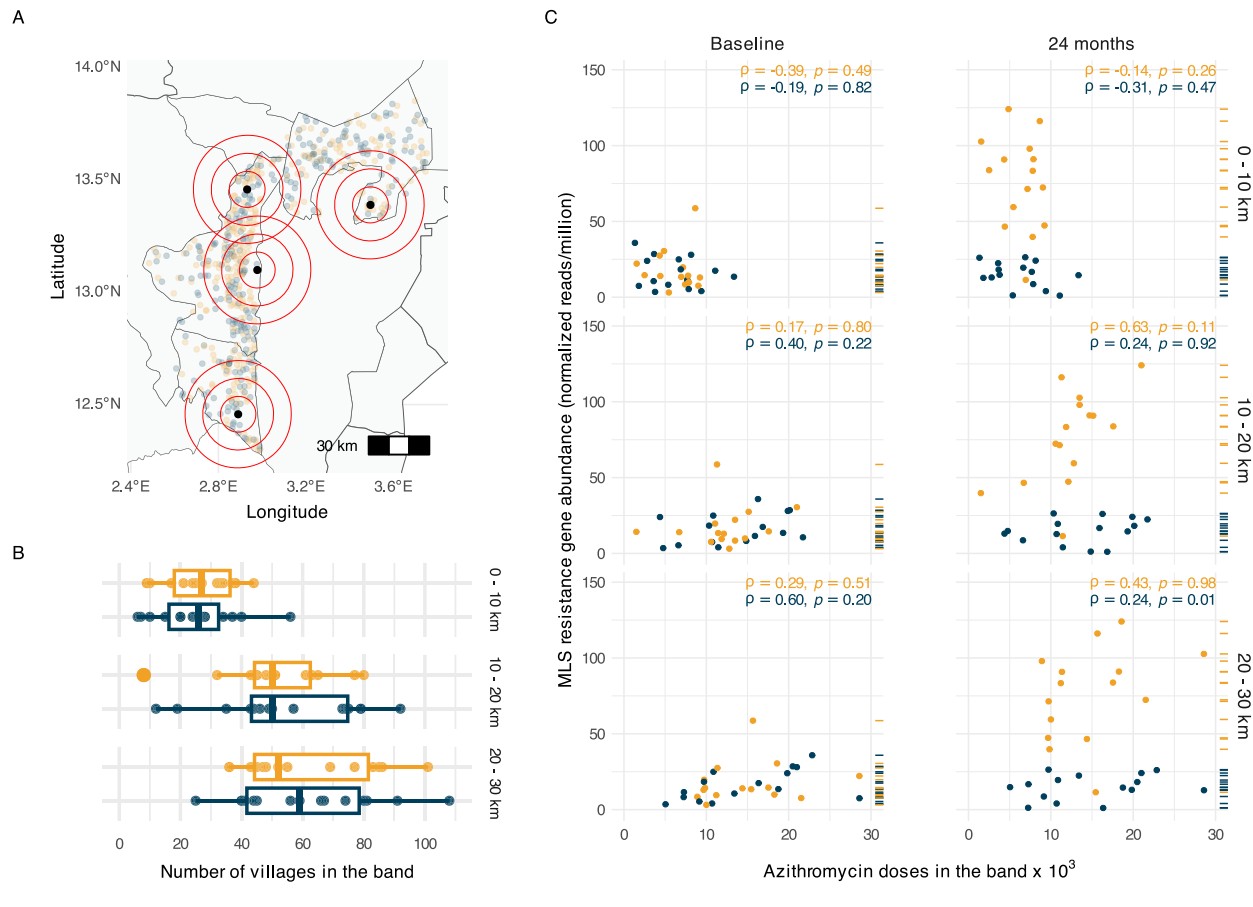

**Fig. 3 | Non-parametric assessment of spillover effects of azithromycin MDA on genotypic resistance to Macrolides–lincosamides–streptogramins (MLS) by 10 km geographic bands. A** Map of the study area indicating the locations of 4 of 30 AMR monitoring villages (black circles) for illustration and all 594 mortality monitoring villages, with azithromycin-treated villages in yellow and placebo-treated villages in blue. Red concentric circles represent distance bands at 0–10 km, 10–20 km, and 20–30 km centered at AMR monitoring villages, used to assess spillover effects of azithromycin treatment intensity. **B** Number of mortality monitoring villages within each distance band from the AMR monitoring villages. Boxplots show the distribution of villages within each band, stratified by treatment assignment of the AMR monitoring villages ($n = 15$ azithromycin-treated villages, $n = 15$ placebo-treated villages). Box boundaries represent the first and third quartiles, the center line represents the median, and whiskers extend from each quartile to the most extreme data point within 1.5 times the interquartile range. As expected from randomization, the distributions are similar for azithromycin-treated and placebo-treated AMR monitoring villages. **C** Association between azithromycin doses administered within each distance band (in thousands) and the normalized abundance of MLS resistance determinants (reads per million) in AMR monitoring villages at baseline (left) and 24 months (right). Each row of scatter plots corresponds to a specific distance band: 0–10 km (top), 10–20 km (middle), and 20–30 km (bottom). Dashes on the right of each panel indicate the normalized abundance of MLS resistance determinants of each village in the panel. Spearman rank-order correlations (ρ) with two-sided *p*-values from permutation tests are shown for both time points. The absence of stronger correlations at closer distances at 24 months suggests no evidence of spillover effects on MLS resistance. Administrative boundaries used in maps were sourced from the United Nations Office for the Coordination of Humanitarian Affairs (OCHA) *Common Operational Datasets – Administrative Boundaries*, accessed via the Humanitarian Data Exchange on 9 April 2024, and are licensed under CC BY-IGO 3.0 (https://data.humdata.org/dataset/cod-ab-ner). Figure created using script https://osf.io/ka68w. Source data are provided as a Source Data file.

Pneumococcus was isolated from 14 of 15 AMR-monitoring villages in the placebo treatment arm and from 13 of 15 in the azithromycin treatment arm, respectively. Consistent with the genotypic resistance findings, no significant associations were observed between geographic treatment intensity of azithromycin and phenotypic macrolide resistance in either placebo-treated villages ($\rho = -0.13$, $P = 1.0$) or azithromycin-treated villages ($\rho = -0.36$, $P = 0.36$; Supplementary Fig. 8A). The negative control exposure analysis using placebo treatment intensity similarly showed no significant associations ($\rho = -0.16$, $P = 0.76$ in placebo-treated villages; $\rho = -0.17$, $P = 0.91$ in azithromycin-treated villages; Supplementary Fig. 8B). These phenotypic resistance data in a clinically relevant respiratory pathogen corroborate the absence of detectable geographic spillover effects observed in the metagenomic analyses.

## Non-parametric analysis of geographic spillovers

To further investigate spatial spillover by isolating the effect of distance from azithromycin-treated mortality monitoring villages, we estimated the association between azithromycin exposure and MLS resistance within discrete distance bands. Specifically, we calculated the cumulative number of azithromycin doses distributed within 10 km, 10–20 km, and 20–30 km of each AMR monitoring community over 24 months (Fig. 3A, B) and assessed their association with MLS resistance using Spearman rank-order correlation; statistical significance was evaluated using a permutation test. In the presence of AMR spillover, we hypothesized that there would be a stronger correlation between azithromycin doses delivered and MLS resistance at smaller distances since closer proximity increases the likelihood of transmission of resistant strains between villages.

The correlations between MLS resistance determinants and geographic treatment intensity within most distance bands were low. These results also suggest no evidence of a significant increase in MLS resistance associated with a higher number of doses distributed in proximate villages (Fig. 3C). Although there was a significant correlation in placebo-treated villages, in the 20–30 km band, this finding should be interpreted cautiously, given that correlations at closer distances were weaker or negative ($\rho = -0.07$ at 0–10 km, $\rho = 0.24$ at 10–20 km), inconsistent with a spillover mechanism where proximal villages would show stronger effects. Moreover, this represents the only significant finding among 12 correlation tests across distance bands, timepoints, and treatment arms, compatible with a chance finding.

### Sensitivity analyses

Gravity models, commonly used to describe human movement and infectious disease dynamics[26], were adapted to assess macrolide resistance spread by using an inverse distance-squared weighted sum of azithromycin doses in mortality monitoring villages as an alternative geographic treatment intensity measure. The results of this sensitivity analysis were similar to the primary analysis (Supplementary Fig. 1), indicating the robustness of the primary analysis to the azithromycin treatment intensity measure used. Additionally, the results were also robust to a leave-one-out analysis, verifying that the findings were not driven by individual AMR monitoring villages (Supplementary Fig. 2).

The pre-specified analysis based on a permutation test could not detect a significant effect of MDA on spillover of macrolide resistance. We conducted an additional, post hoc analysis to estimate the fold-change in resistance genes corresponding with an increase of azithromycin doses in a 10 km radius surrounding a village. Unlike the conditional permutation test, inference no longer relies exclusively on the randomization, but it provides a complementary assessment of the precision around the observed null effect. In log-linear models adjusted for baseline macrolide resistance load, as well as for azithromycin doses distributed within the 10–20 km and 20–30 km distance bands, we estimated that an increase of 5,000 azithromycin doses within a 10 km radius (corresponding to the median value of observed doses) would result in a fold change in macrolide resistance genes of 0.7 (95% CI: 0.2–1.9) in placebo-treated villages and 1.0 (95% CI: 0.4–2.3) in azithromycin-treated villages. Point estimates are consistent with the randomization-based inference (no spillover) and the upper limits of the intervals show that modest increases in resistance genes, up to a 2.3-fold increase, are unlikely but compatible with the data given the relatively small number of 15 monitoring villages in each analysis.

We additionally assessed the relationship between the measure of geographic treatment intensity and independent measures of population density and distance to primary health post to help provide additional context for the results. High-resolution population estimates[27] were associated with geographic treatment intensity and MLS resistance (Supplementary Fig. 4), illustrating that observational analyses that fail to account for population density could be confounded. Re-fitting the log-linear models to estimate fold-changes, adjusting for population density in addition to baseline macrolide resistance load and azithromycin doses within the 10–20 km and 20–30 km distance bands, led to estimates of 0.3 (95% CI 0.1–1.3) in placebo-treated villages and 0.7 (95% CI 0.2–1.9) in azithromycin-treated villages. Proximity to the nearest health center[28], which could influence access to antibiotics and selection pressure, was not associated with either geographic treatment intensity or AMR prevalence (Supplementary Fig. 5).

Finally, we extended our analysis to investigate potential spillover effects on genotypic resistance to other classes of antibiotics beyond MLS. No significant relationship was observed between geographic treatment intensity and non-macrolide antimicrobial resistance determinants at either baseline or 24 months (Supplementary Fig. 6).

These findings align with previously reported results, which found no significant between-group differences in resistance for non-macrolide antimicrobial classes at 24 months[8].

## Discussion

This study found no evidence of between-village geographical spillover of macrolide AMR to nearby untreated villages following large-scale, twice-annual azithromycin MDA to children aged 1–59 months old in Niger. Genetic load of MLS resistance in placebo-treated monitoring villages remained at baseline levels after 24 months of intervention and had no association with the number of azithromycin doses administered in nearby treated villages. These findings were consistent across both, genotypic resistance measured through metagenomic sequencing and phenotypic macrolide resistance in *S. pneumoniae*, further strengthening the evidence. These results do not support the hypothesis that azithromycin MDA-induced selection of AMR extends beyond treated villages at the geographic scales evaluated. Spillover effects at smaller spatial scales, such as within-household or within-village, could be present[20,29], but substantial, unmeasured indirect effects of mass drug administration on resistance are unlikely beyond the effects observed within treated communities over a 24-month period (four MDA treatments). Our findings suggest that current sentinel village AMR monitoring strategies, which do not explicitly account for between-village spillover effects, likely captured the AMR selection attributable to MDA during the initial years of program implementation in this setting. Internally consistent results across multiple analytical methods reinforce our inference and demonstrate that effects on AMR selection estimated in the original trial analysis are unlikely to be biased by between-cluster spillover. This analysis provides new insights into the spatial scale of azithromycin treatment variation and AMR selection following MDA and contributes to evidence-based decision-making regarding the use of azithromycin MDA for reducing childhood mortality in high-risk settings.

Unlike conventional randomized trials where outcomes are measured among treated and untreated units, in this analysis the MDA treatments used to quantify spillover effects were delivered to different clusters (594 villages) than those used to measure AMR outcomes (30 monitoring villages), a generalization of the randomized controlled trial[30]. A random sample of 30 villages was separately randomized to avoid potential effects of intensive sample collection on the primary mortality outcome while allowing us to isolate spillover effects. Cluster randomization over 594 villages created variation in geographic treatment intensity across the independently sampled 30 AMR monitoring villages, and permuting the random assignments using a conditional permutation test allows for an exact, non-parametric test for the presence of spillover effects between villages[22,23]. Along with failure to reject the null under the conditional permutation test, the separate finding of a similar correlation between geographic treatment intensity and AMR, regardless of whether geographic treatment intensity was estimated using azithromycin or placebo doses (Fig. 2B, D), further supports the view that the observed correlation is due to factors other than spillover. Robustness checks that adjusted for local population density and distance to health center did not change our inference and reinforced the conclusion that geographic treatment intensity did not independently influence AMR in this study.

The analysis has several caveats. Although simple random sampling resulted in some geographic clustering that reflects underlying settlement patterns, the 30 monitoring villages spanned the full range of observed treatment intensity and showed similar distributions of geographic factors such as distance to health posts (Supplementary Fig. 10) compared to the 594 mortality-monitoring villages. Despite adequate spatial coverage, the relatively small number of AMR monitoring villages ($N = 15$ per arm), with 10 children sampled at each village, may limit statistical power. The upper bound of our post-hoc

confidence intervals (up to 1.9-fold increase) suggests we cannot definitively rule out modest spillover effects. Appropriate power calculations for geographic spillover with spatial clustering remain an open methodological challenge, requiring complex simulations to account for settlement patterns, between-village correlations, and distance-decay functions. Future studies with larger monitoring samples and methodological advances in power estimation for spatially-structured spillover effects would strengthen inference about small-magnitude geographic spillover effects. Another potential limitation is the choice of spillover measure: we assume the potential for AMR spillover is proportional to the absolute number of azithromycin doses distributed rather than the proportion of the population treated. This approach was chosen because the trial's restriction to non-urban villages of 200–2000 residents, with exclusion of larger settlements, makes population-based proportions unreliable due to incomplete sampling and edge effects. The distance decay is also assumed to be uniform across the study area, without accounting for potential variations in ease of connectivity and travel time between villages or geographic factors such as terrain and remoteness. This assumption may oversimplify some spatial relationships, and the incorporation of high-resolution data on mobility patterns and social networks could strengthen the sensitivity. Although we examined both genotypic and phenotypic resistance, these measures serve as indicators of resistance potential rather than direct evidence of clinical treatment failure. Finally, our findings may not be directly generalizable to trachoma control programs, which differ in two key aspects: trachoma MDA typically targets entire communities rather than only children aged 1–59 months, and treatment frequency is generally annual rather than biannual. While children aged 1–59 months may not travel independently between villages, villages in this setting regularly share community infrastructure including schools, markets, and health posts, providing opportunities for contact and potential transmission of resistant bacteria between children from different villages. The lack of observed spillover despite these opportunities may reflect insufficient contact intensity or duration for sustained between-village transmission, limited persistence of resistant strains without ongoing selective pressure, or rapid strain turnover in young children's gut microbiomes. The broader population coverage and higher mobility of adults in trachoma programs could potentially alter spillover dynamics through increased treatment intensity and expanded networks of transmission. This study provides insights into local geographic spillover effects over a 24-month period, but azithromycin MDA has continued in the Dosso region for several years beyond this study[31], and the use of azithromycin MDA in trachoma control programs regularly spans several years. Longer-term monitoring of AMR trends in regions with repeated rounds of MDA would help measure cumulative effects on resistance over time, including potential geographic spillover effects. Future analyses of large-scale, azithromycin MDA trials in the region[3,4,32] should enable tests for spillover effects over longer periods and at smaller spatial scales.

In summary, there was no evidence of between-village spillover of macrolide resistance following azithromycin MDA, suggesting that AMR effects are largely localized to treated villages without substantial spreading to neighboring areas at the spatial and temporal scales evaluated. This study highlights the value of randomized trial designs to test for spillover effects of large-scale public health interventions even if outcome monitoring is focused in a subsample of units. Future studies should continue to explore the longer-term impacts of repeated azithromycin MDA on selection and possible geographic spillovers of AMR.

## Methods

This is a secondary analysis based on the MORDOR mortality trial (NCT02047981), a cluster-randomized trial that evaluated the impact of twice-yearly mass distribution of oral administration of azithromycin on childhood mortality, and its sister trial, MORDOR Morbidity trial (NCT02048007). The geographic spillover analysis plan was pre-specified (https://osf.io/xyqnr) after the main trial results were published but prior to conducting spillover analyses.

### Trial oversight

The trial protocol was reviewed and approved by the Committee for Human Research at the University of California, San Francisco (protocol #10-01036) and the Ethical Committee of the Niger Ministry of Public Health. The trial was monitored by an independent Data and Safety Monitoring Committee and was conducted in accordance with the Declaration of Helsinki. Due to low literacy in Niger, trial oversight committees approved the use of oral consent from guardians of children before administration of treatment and each sample collection. No incentives were offered for participation in this study.

### Study population

The trial was conducted in the Boboye, Loga and Falmey departments in Dosso region of Niger (Fig. 1A). All non-urban villages with an estimated population of 200–2000 people in the study area were eligible to be included in the trial. Eligible villages were randomized to receive MDA with azithromycin or placebo to monitor the impact of azithromycin MDA on childhood mortality. Additionally, a random sample of 30 other villages from the same pool of eligible villages were separately randomized to receive MDA of azithromycin or placebo to monitor the impact of MDA on AMR (Fig. 1B). All children aged 1–59 months in enrolled villages received the assigned treatment, azithromycin or matching placebo, twice annually for 2 years. Baseline AMR measurements were collected in the AMR-monitoring villages prior to the first MDA distribution, starting in December 2014, and follow-up AMR measurements were collected at 24 months, six months after the fourth MDA distribution.

### Geographic treatment intensity

In this context, geographic treatment intensity represents the potential for geographic between-village spillover of AMR resulting from azithromycin treatment in nearby clusters. A geographic treatment intensity variable was computed based on the cumulative number of doses of azithromycin distributed as part of the mortality monitoring trial. Geographic treatment intensity was estimated as an inverse distance weighted sum of the number of doses of azithromycin distributed at each AMR-monitoring location to allow for higher weights for nearby treated villages while accounting for diminishing influence with increasing distance. If the number of azithromycin doses distributed at locations $x_1,...,x_n$ are $v_1,...,v_n$ respectively, then the geographic treatment intensity at a location $u$ is $g(u) = \sum w_i v_i$ where the weights are the inverse $p^{th}$ powers of distance, $w_i = 1/d(u,x_i)^p$ where $d(u,x_i) = ||u-x_i||$ is the great-circle distance from $u$ to $x_i$. As the power is reduced, the smoothness of the interpolated spatial distribution increases.

The primary analysis was done using a power of 1, and a sensitivity analysis was done using a power of 2. A power of 1 in the inverse distance weighting calculation was selected for the primary analysis as it provides a balance between capturing the influence of nearby treated villages while maintaining contributions from more distant villages, analogous to Shepard's method[33] but applied to weighted sums rather than interpolated averages. This linear decay function maintains a relatively high influence within the first several kilometers before diminishing at greater distances, which aligns with the expected pattern of antimicrobial resistance dissemination across geographic areas[34]. Gravity models are widely used to describe human movement across spatial scales, particularly in settings where high-resolution data on mobility patterns, transportation routes, or social networks may be limited. They have recently been applied to model population mixing and mobility in infectious disease transmission dynamics[26,35]. We

adapted the principles of gravity models to evaluate the spatial dissemination of macrolide resistance by assessing the inverse distance-squared weighted sum of the cumulative number of azithromycin doses distributed in mortality monitoring villages as a sensitivity analysis.

In instances when treated village distances fell below 1 km from monitoring villages, distances were truncated at 1 km to avoid inordinately large weights in the calculation. The geographic treatment intensity variable was rescaled to range from 0 to 1 by dividing by its maximum value. AMR monitoring villages were assigned the geographic treatment intensity value at their geographic centroid.

In this geographical treatment intensity calculation, we assume that the potential for AMR spillover is proportional to the absolute number of azithromycin doses distributed rather than the proportion of the population treated. Additionally, this approach does not account for the possibility that larger AMR monitoring villages might experience more spillover events due to increased human mobility and bacterial transmission or, conversely, that smaller AMR monitoring villages might exhibit more measurable spillover effects due to lower background resistance levels. However, there was no clear association between AMR-monitoring village size and either geographic treatment intensity (Supplementary Fig. 3) or MLS resistance gene abundance (Supplementary Fig. 7), suggesting that village size did not systematically influence the observed outcomes. Furthermore, we assume that geographic distance alone dictates the strength of spillover effects, without incorporating other potential modifying factors such as population density, healthcare access, or environmental pathways of resistance transmission.

## Village load of antimicrobial resistance genes

Rectal swabs were collected from a random sample of 10 children aged 1 to 59 months in each AMR monitoring village at baseline and at 24 months. DNA was extracted from each sample, and its concentration was quantified and normalized for sequencing library preparation. Libraries were sequenced with 150-nucleotide paired-end reads. Reads underwent three rounds of human sequence removal, quality filtering, and taxonomic classification before non-human reads were aligned to the MEGARes database to identify antibiotic-resistance determinants (ARDs) with a gene fraction greater than 80%. A sample was determined to be macrolide resistant if at least one ARD in the MLS class was detected. Microbiome sequencing reads can be accessed at https://microbiomedb.org/mbio/app. Detailed laboratory protocols for sample processing, DNA extraction, library preparation, and sequencing are provided in detail in previously published studies[8,36]. The normalized abundance of these determinants (measured as reads per million) was calculated for each sample[8,9]. Community-level averages of MLS resistance determinants were used as the primary outcome measure. Community-level means were highly correlated with the proportion of children with detectable resistance within villages (Supplementary Fig. 11).

Similarly, normalized abundance of resistance determinants was estimated for other classes of antibiotics, namely, aminocoumarins, aminoglycosides, bacitracin, beta-lactams, elfamycins, fluoroquinolones, fosfomycin, glycopeptides, metronidazole, phenicol, rifampin, sulfonamides, tetracyclines, trimethoprim, and multidrug resistance.

Nasopharyngeal swabs were collected from 10 randomly selected children per village at 24 months, stored in skim milk-tryptone-glucose-glycerin (STGG) medium, and shipped to the University of California, San Francisco for processing. Pneumococcus isolation and antimicrobial susceptibility testing were performed at ARUP, a CLIA-certified laboratory, using standard broth microdilution methods and Clinical and Laboratory Standards Institute breakpoints; further details are provided in the supplementary appendix of Doan et al.[8]. The proportion of erythromycin-resistant isolates was calculated for each village where pneumococcus was successfully isolated. Associations between geographic treatment intensity and the proportion of macrolide-resistant *S. pneumoniae* were evaluated using Spearman rank-order correlations, with statistical significance assessed using conditional permutation tests as described above.

## Spillover effect assessment

To assess the potential spillover effects of azithromycin MDA on MLS resistance, the association between azithromycin treatment intensity and the normalized abundance of MLS resistance determinants was evaluated. The primary analysis focused on placebo-treated monitoring villages, testing the hypothesis that resistance in these villages was associated with the geographic treatment intensity of azithromycin in surrounding villages. Additionally, the association within azithromycin-treated monitoring villages was examined to explore whether a similar relationship existed despite the direct effects of treatment.

Spearman rank-order correlations were used to evaluate these associations, stratified by whether the monitoring villages received placebo or azithromycin. Analyses were conducted at baseline and 24 months post-MDA to account for temporal dynamics. To ensure the robustness of the correlations estimated, a leave-one-out analysis was conducted, sequentially excluding each AMR monitoring village from the dataset to evaluate the influence of individual villages on the overall results. In addition to MLS resistance, we also assessed resistance determinants for other antibiotic classes to detect broader spillover effects.

To test the statistical significance of observed correlations between geographic treatment intensity and MLS resistance, we used a non-parametric, conditional permutation test to detect the presence of spillover effects between villages[22,23]. In each permutation, the treatment labels of the 594 mortality monitoring villages were re-randomized, and the geographic treatment intensity variable was re-estimated for the AMR monitoring villages. AMR monitoring village treatments were held fixed in the permutation tests, creating a distribution for the correlation between the monitoring village AMR load and surrounding geographic treatment intensity under the null hypothesis of no between-village spillover effects[22,23]. The null distribution for all test statistics was generated from 1000 permutations. Observed correlations were compared to the null distribution to determine two-sided *p*-values, with significance assessed at an alpha of 0.05.

## Negative control exposure analysis

The placebo treatment in mortality-monitoring villages served as a negative control exposure to assess robustness. We repeated the geographic spillover analysis, substituting the cumulative number of placebo doses for azithromycin doses when estimating the geographic treatment intensity, using the same inverse distance-weighted approach and identical testing procedures. Randomization of treatment assignments in the mortality monitoring villages balanced all potential confounders—both measured and unmeasured, including spatially structured factors—across treatment arms. The geographic treatment intensity of placebo is identically structured to the geographic treatment intensity of azithromycin, yet has no causal effect on AMR, making it an ideal negative control exposure that captures all sources of confounding that affect both treatment intensity measures similarly. Any observed associations between the geographic treatment intensity of placebo and AMR load could be attributed to confounders unrelated to azithromycin exposure.

## Non-parametric treatment measure

To separate the effect of distance and the number of doses, a non-parametric distance-based analysis was conducted. Considering each AMR monitoring community as the center, concentric non-

overlapping distance bands set at 0–10 km, 10–20 km, and 20–30 km were established. The cumulative number of azithromycin doses administered within each band was used as an alternative azithromycin treatment intensity estimate within each ring. The number of azithromycin doses within each distance band was examined as a predictor of MLS resistance determinants in the AMR monitoring community at baseline and 24 months. Associations between azithromycin doses in each distance band and MLS resistance determinants were estimated using Spearman rank-order correlations and their significance was evaluated using a permutation test.

### Additional sensitivity analyses

The pre-specified analysis focused on the randomization in the design and conditional permutation test for inference. After review of the main results and negative control analysis, we conducted three additional model-based sensitivity analyses as robustness checks (not pre-specified). First, we fitted log-linear models relating normalized macrolide resistance gene load to the number of azithromycin doses within 0–10 km, 10–20 km, and 20–30 km of each village, adjusting for baseline macrolide resistance levels. From these models, we estimated the fold-change in macrolide resistance gene abundance associated with an increase of 5,000 doses within 10 km. We fit separate models for placebo-treated and azithromycin-treated monitoring villages, and estimated model-based 95% confidence intervals and P-values for the fold-change. Second, we evaluated the association between two geographic characteristics that could be correlated with our measure of geographic treatment intensity and load of antimicrobial resistance: local population density and distance to the nearest health center. We characterized local population density using data from the Meta High Resolution Settlement Layer[27] and summarized the number of children under 5 years within the distance rings of 0–10 km, 10–20 km, and 20–30 km of each AMR-monitoring village. We also estimated distance to the nearest Centres de Santé Integré facility, a primary health center that is equipped to provide basic inpatient and outpatient care. After determining that population density was associated with both geographic treatment intensity and macrolide resistance, we included it as an additional covariate in the log-linear model.

### Statistics & Reproducibility

The 594 mortality monitoring villages and 30 AMR monitoring villages were independently randomized using computer-generated assignments to azithromycin or placebo. Sample size for the AMR monitoring trial (15 villages per arm) was determined to detect differences in the primary AMR outcome of direct treatment effects; no statistical method was used to predetermine sample size for the geographic spillover analysis. No villages or samples were excluded from the geographic spillover analyses; all 30 AMR monitoring villages were successfully analyzed at both timepoints, and all rectal swab samples were successfully sequenced. Laboratory personnel conducting sample processing, sequencing, and AMR quantification were blinded to village treatment assignments; statistical analysts were not blinded as knowledge of spatial treatment patterns was necessary for analysis. The geographic spillover analysis plan was pre-specified (https://osf.io/xyqnr) after main trial publication but prior to conducting spillover analyses; three model-based sensitivity analyses were conducted post hoc as robustness checks. Analyses used R statistical software (version 4.4.0, 2024-04-24 "Puppy Cup"). Key packages used are: broom 1.0.11, cowplot 1.2.0, ggpubr 0.6.2, ggspatial 1.1.10, ggtext 0.1.2, here 1.0.2, lwgeom 0.2-14, patchwork 1.3.2, raster 3.6-32, sf 1.0-23, tidyverse 2.0.0, units 1.0-0. Full list available at https://osf.io/x4e3n and complete replication code available at https://osf.io/qxtec/.

### Reporting summary

Further information on research design is available in the Nature Portfolio Reporting Summary linked to this article.

## Data availability

The trial data used in this analysis have been deposited in Open Science Framework (https://doi.org/10.17605/OSF.IO/BMJD3)[37] Village geographic coordinates data are protected and are not available to protect participant confidentiality. The microbial sequencing reads have been deposited with the NCBI Sequence Read Archive under BioProject no. PRJNA1356862. Source data are provided with this paper. Administrative boundaries used in maps were sourced from United Nations Office for the Coordination of Humanitarian Affairs' Common Operational Datasets are hosted on Humanitarian Data Exchange (accessed at https://data.humdata.org/dataset/cod-ab-ner)[38], and are licensed under CC BY-IGO 3.0. Population estimates were obtained from High Resolution Settlement Layer by Meta's Data for Good (accessed at https://data.humdata.org/dataset/highresolutionpopulationdensitymaps-ner)[27]. Replication code for all analyses is also available through the Open Science Framework (https://osf.io/rc9qj/files/github). Source data are provided with this paper.

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

## Acknowledgements

The MORDOR trial was supported by a grant from the Gates Foundation (OP1032340 to TML). This work was additionally supported in part by the National Institute of Allergy and Infectious Diseases (R01AI175250 to KSO and TML, R01AI158884 to BFA, R01AI166671 to BFA, R35GM147702 to SB) and the Centers for Disease Control and Prevention (CDC U01CK000590 to AS and SB, as part of the Modeling Infectious Diseases in Healthcare Network). Pfizer provided both the azithromycin and the placebo oral suspensions.

## Author contributions

Following CRediT taxonomy, conceptualization (A.S., T.M.L., B.F.A.), data curation (A.H., T.D., B.F.A.), formal analysis (A.S.), funding acquisition (B.F.A., T.M.L.), investigation (A.S., T.D., A.H., B.F.A.), methodology (A.S., M.L., T.C.P., T.M.L., B.F.A.), project administration (A.M.A., R.M., A.A.), software (A.S., B.F.A.), supervision (T.M.L., B.F.A.), validation (A.S., B.F.A.), visualization (A.S., B.F.A.), Writing—Original Draft Preparation (A.S., B.F.A.), Writing—Review & Editing (A.S., A.M.A., R.M., A.A., M.L., S.B., K.S.O., T.C.P., A.H., T.D., J.D.K., T.M.L., B.F.A.).

## Competing interests

The authors declare no competing interests.

## Additional information

[1]Francis I Proctor Foundation, University of California San Francisco, San Francisco, USA. [2]Department of Epidemiology & Biostatistics, University of California San Francisco, San Francisco, USA. [3]Centre de Recherche et Interventions en Santé Publique, Birni N'Gaoure, Niger. [4]Programme National de Santé Oculaire, Niamey, Niger. [5]Center for Communicable Disease Dynamics, Department of Epidemiology, Harvard T.H. Chan School of Public Health, Boston, USA. [6]Department of Immunology and Infectious Diseases, Harvard T.H. Chan School of Public Health, Boston, USA. [7]Department of Medicine, University of California San Francisco, San Francisco, USA. [8]Department of Ophthalmology, University of California San Francisco, San Francisco, USA. [9]Institute for Global Health Sciences, University of California San Francisco, San Francisco, USA. ✉e-mail: ben.arnold@ucsf.edu

