## [Transparent Peer Review file · Nature Communications]

Geographic spillover of antimicrobial resistance from mass distribution of azithromycin

Corresponding Author: Professor Benjamin Arnold

Version 0:

Reviewer comments:

Reviewer #1

(Remarks to the Author)

This manuscript has a number of carefully constructed analyses to investigate possible increases in AMR in villages near those that were given mass administration of azithromycin. Sensitivity analyses are extensive, and usefully include a negative control exposure, and a leave-one-village-out analysis (important, given there were only 15 untreated villages).

Questions and comments:

In what sense was this 'a pre-specified secondary analysis of the MORDOR trial in Niger'? Pre-specified before the trial started, or after the main trial results were out and the differential effects by country were noted?

Top p.10: Great not capitalized in great circle

Would be good to see the most simple-minded analysis, which according to the figures, would also be not at all significant: change (or 24mo regressed on baseline) difference between the 15 control and 15 intervention monitoring villages.

Likewise—could this be done not with the gene abundance, but with actual numbers of children (or proportions) with resistance? Is it possible that a village with 50 reads/million, for example, only had one such child? Could 2 children with resistance in one village account for more reads than 5 in another?

Given the preceding concern, it's good the analyses employ nonparametrically-derived p-values.

I didn't see anything about nasopharyngeal swabs—what about their results?

(Remarks on code availability)

There is certainly enough information here to reproduce the results.

Reviewer #2

(Remarks to the Author)

(Remarks on code availability)

Reviewer #3

(Remarks to the Author)

This is an important secondary analysis of the MORDOR mortality trial examining the potential geographic AMR spillover of MDA azithromycin for reducing childhood mortality. The methodology carefully lays out various statistical approaches to overcome limitations inherent in the original study design and data collection. The findings focus on an additional, small random sample of 30 villages, with 15 villages receiving MDA of azithromycin and 15 villages receiving placebo, amidst the 594 surrounding villages where the mortality benefit of MDA azithromycin was assessed. This monitoring occurred only in Niger, where the most pronounced benefits of lowering childhood mortality were found, in the original MORDOR study that also included study sites in Malawi and Tanzania.

The biological plausibility of intra-community-level transmission of AMR determinants of resistance to azithromycin is not in question, as this clearly has been documented in another study of MORDOR data. It is helpful to have documented that inter-community spread of AMR may not have been a concern in the conduct of this MDA azithromycin trial, where the treated population were children aged 1-59 months. The relevant take-away questions, of course, are why. Is it because the factors that spread macrolide resistance are not present between villages or the sample size for AMR monitoring was too small to detect a difference?

It would be clearer to use “genotypic resistance” related to this study’s findings, to avoid confusion, as this study did not address phenotypic resistance. As you know, in some cases, phenotypic resistance to macrolides may be observed when genotypic resistance is not detected by standard genetic tests.

Introduction (page 2): The discussion of prior MORDOR data tracking macrolide resistant determinants in azithromycin-treated villages, both among children in the study vs. others, would offer useful context. It also would better anticipate the much needed section in the Discussion as to why macrolide resistant determinants were not detected to have spread across villages in this study. Policymakers should not come away from this study imagining that somehow it is not biologically plausible that AMR spreads via humans, food products, or the environment.

Measure of azithromycin treatment intensity (page 4): “We quantified geographic treatment intensity at any point in the study region as the cumulative number of doses of azithromycin distributed in mortality-monitoring villages over 24 months (Figure 1C), weighted by the inverse of the distance to that point. We used an inverse distance weighting approach with a linear decay function to balance the influence of nearby and distant treated villages, aligning with the expected antimicrobial resistance spread.”

1. How does the “inverse distance weighting approach” account for factors that a more detailed investigation of transportation routes, mobility patterns, social exchange networks, administrative boundaries, or geographical barriers (waterways or mountainous terrain) might have? As the authors note in the Introduction, population mobility can be a fundamental driver of spread.
2. What overlap was there with MDA for trachoma in the Niger study region, and might that have influenced the measure of azithromycin treatment intensity? Is MDA for trachoma a potential confounder to observations made in this secondary analysis?

Methods – Several additional methodological details would help the reader evaluate the study:

1. With only 15 placebo villages, what difference was the study powered to detect? Although demonstration of evidence of genotypic resistance in treated communities is helpful context, the focus here should be on the placebo communities.
2. The authors should provide more details about the MLS genotypic testing. Does this test for all known genotypic mutations leading to macrolide resistance? What level of detection is considered significant? On page 10, gene fraction >80% is noted – is this standard?

Results—Random sample of 30 additional villages (page 3): It would be helpful to the reader to understand why AMR monitoring was not embedded as a subset, as opposed to an additional sampling, of the villages. Figure 1B suggests that the separate random sample of 30 villages are heavily concentrated in one part of the Dosso region of Niger.

1. Should the reader know whether this geographical concentration correspond to any factors (such as the aforementioned concerns with distance alone) that might have influenced the true dynamics of resistance spillover?
2. Inverse distance weighting performs less well when sample points cluster. Is this a methodological concern here (or not) given the apparent geographical clustering of the randomly selected villages in the AMR monitoring sample within the Dosso region?

Results—Spillover effects of azithromycin MDA to nearby villages (page 4): Could the authors address potential confounders that likely will arise in the mind of readers?

1. Does the fact that MLS resistance determinants remained low and at baseline levels in placebo-treated AMR monitoring villages at 24 months provide sufficient assurances (or just a higher baseline) from confounding factors, such as the frequent administration of antibiotics for childhood ailments in these settings?
2. Macrolide antibiotics are one of the more commonly used families of antibiotics in food animal production in Sub-Saharan Africa (<https://pmc.ncbi.nlm.nih.gov/articles/PMC10830768/>). Could this have potentially influenced observations made in the study? More intensive animal food production systems may use antibiotics to a greater degree. Would these also be randomly distributed throughout the Dosso region?

In this section, it is also noted that: “A modest correlation between geographic treatment intensity and MLS resistance in azithromycin-treated villages was observed at baseline ($p=0.30$) and remained similar at 24 months ($p=0.32$), though neither was significant under a conditional permutation test ($P=0.21$ at baseline, $p=0.95$ at 24 months).” Actually, none of the p -

values are significant at the $P < 0.05$ level. How do we know that the sensitivity of this measure to detect AMR spillover is sufficient for doing so?

“Metagenomic deep sequencing was performed on pooled rectal swab samples from 40 children per village...” How might have the pooling of rectal swab samples affected the validity that might have been resulted from an analysis of the individual samples? Were these samples transported to local laboratories or to a more distant central laboratory for processing? Were there any systematic biases in metagenomic sequencing that might be introduced from transport and processing of these samples that required correction (<https://pmc.ncbi.nlm.nih.gov/articles/PMC6739870/>)?

Results—Negative control exposure analysis (page 4): “Using the same methods used to estimate azithromycin geographic treatment intensity, but with the cumulative number of placebo doses distributed instead of the cumulative azithromycin doses, the negative control exposure analysis was designed to detect spurious associations arising from unmeasured confounding, or other non-causal sources of variation in AMR that are unrelated to azithromycin MDA [22,23].” Does not negative control exposure analysis only adjust for some types of unmeasured confounding, but not other unmeasured confounding (e.g., macrolides used in livestock raising in some villages but not others)?

Results—Non-parametric analysis of geographic spillovers (page 5): “In the presence of AMR spillover, we hypothesized that there would be a stronger correlation between azithromycin doses delivered and MLS resistance at smaller distances since closer proximity increases the likelihood of transmission of resistant strains between villages. The correlations between MLS resistance determinants and geographic treatment intensity within most distance bands were low.” The discrete distance bands were within 10 km, 10-20 km, and 20-30 km of each AMR monitoring community over 24 months. Does this lack of a gradient suggest the insensitivity of the inverse distance weighting approach in capturing this?

Results—Sensitivity analyses (page 5): Relying on physical distance, inverse distance weighting might not adequately capture actual functional connectivity. How might network-based models or the gravity model discussed offset these limitations? The “gravity models...were adapted to assess macrolide resistance spread by using an inverse distance-squared weighted sum of azithromycin doses in mortality monitoring villages as an alternative geographic treatment intensity measure.” How is this gravity model different from the inverse distance weighting?

1. Rather than examine AMR spillover in proportion to the absolute number of azithromycin doses distributed, might an alternative approach have considered the proportion of the population treated, or even better, the proportion of the treated population that might transit between villages?
2. Given that those treated with MDA azithromycin were aged 1-59 months, might this segment of the population not be particularly mobile between villages at that age? Might that explain the finding of no AMR spillover between villages? If that is the case, might it be important to note that the findings in this secondary analysis of MORDOR may not be generalizable to, say, MDA trachoma in the general population?

Results – Figure 3C. The authors dismiss a statistically significant finding at the 20-30 km radius for the placebo groups because it was not consistent with their hypothesis that there would be strong correlations at closer distances. Might this finding not simply undermine the hypothesis, warranting more careful interpretation of the findings and subsequent Discussion?

Discussion (pages 6-8): Some of the foregoing issues may warrant further explanation in the Discussion section, including:

- 1) limited sample size of AMR monitored villages;
- 2) reasons for lack of finding AMR spillover between villages in this context, including azithromycin-treated group being 1-59 months of age;
- 3) generalizability (or not) given MDA trachoma involves the entire population, notably age segments of the population more mobile across villages.

The Discussion notes that the “findings suggest that current AMR monitoring strategies in sentinel villages...are likely sufficient.” Whether this follows from the findings may depend on responses above – was the study sufficiently powered? Would more targeted sentinel communities, based on proximity and human mobility, actually be a better method, compared to random sampling?

(Remarks on code availability)

Reviewer #4

(Remarks to the Author)

(Remarks on code availability)

Version 1:

Reviewer comments:

Reviewer #1

(Remarks to the Author)

Great to see the additional by-village change analyses.

(Remarks on code availability)

Reviewer #3

(Remarks to the Author)

We appreciate the authors' efforts to address the methodological concerns raised about this paper. Most were addressed with useful caveats and acknowledged limitations in the text. The absence of spillover is not biologically plausible, in general, but only possible in specific circumstances. The question is what this study sheds insight regarding what are those circumstances and whether current AMR monitoring strategies are sufficient, compared to better or more comprehensive approaches to monitoring. The difficulty in assessing whether the study is sufficiently powered, the geographical clustering, the lack of data on actual ties between communities (mobility of people), and the lack of information about potential confounders like the use of antibiotics in local livestock raising (which unfortunately in West Africa is prevalent among both small farmers and intensive farming operations —<https://onlinelibrary.wiley.com/doi/full/10.1002/agr.21770>) make determining what those insights are challenging. Taking into account the study's limitations, it might be useful to rethink whether one should be calling for continuing just more of the same AMR monitoring strategies or improved vigilance ("Our findings support current AMR monitoring strategies in sentinel villages, as we found no evidence of geographic spillover, which could cause these surveillance strategies to underestimate AMR selection during the initial years of MDA implementation..."). Given what is at stake regarding AMR and child mortality, the absence of evidence may not be evidence of absence, and the paper's findings should be framed and interpreted accordingly.

(Remarks on code availability)

Reviewer #4

(Remarks to the Author)

(Remarks on code availability)

Reviewer #1 (Remarks to the Author)

This manuscript has a number of carefully constructed analyses to investigate possible increases in AMR in villages near those that were given mass administration of azithromycin. Sensitivity analyses are extensive, and usefully include a negative control exposure, and a leave-one-village-out analysis (important, given there were only 15 untreated villages).

Questions and comments:

1. In what sense was this 'a pre-specified secondary analysis of the MORDOR trial in Niger'? Pre-specified before the trial started, or after the main trial results were out and the differential effects by country were noted?

Response: This is a good point regarding the timing of our analysis plan vis-à-vis the main results. The analysis plan was pre-specified on the Open Science Framework after the main MORDOR trial results were published, but before conducting any analyses of geographic spillover effects. In response to this comment, we have revised the methods to clarify the timing.

Lines 440-441 (Methods):

"The geographic spillover analysis plan was pre-specified (<https://osf.io/xyqnr>) after the main trial results were published but prior to conducting spillover analyses."

2. Top p.10: Great not capitalized in great circle

Response: Thanks. We have corrected the typographical error.

Line 479 (Methods):

"...where $d(u, x_i) = \|u - x_i\|$ is the great-circle distance from u to x_i ."

3. Would be good to see the most simple-minded analysis, which according to the figures, would also be not at all significant: change (or 24mo regressed on baseline) difference between the 15 control and 15 intervention monitoring villages.

Response: We agree that this analysis provides an intuitive and complementary perspective on the results. We added Supplementary Figure 9, which presents the analysis with the change in MLS resistance gene abundance from baseline to 24 months for each monitoring village as the outcome. Panel A shows the individual village trajectories, illustrating that azithromycin-treated villages experienced substantial increases while placebo-treated villages remained stable. Panels B and C examine whether these changes correlate with geographic treatment intensity of azithromycin and placebo, respectively. As the reviewer anticipated, neither analysis shows significant associations (treatment intensity of azithromycin: $\rho = -0.09$, $p = 0.95$ in placebo-treated villages and $\rho = 0.12$, $p = 0.64$ in azithromycin-treated villages).

Lines 171-175 (Results):

"Additionally, when examining the change in MLS resistance from baseline to 24 months, no significant association was observed with geographic treatment intensity in either placebo ($\rho = -0.09$, $P = 0.95$) or azithromycin-treated villages ($\rho = 0.12$, $P = 0.64$, Supplementary Figure 9B), consistent with the absence of evidence for geographic spillover effects."

Lines 192-196 (Results):

"The change in MLS resistance from baseline to 24 months similarly showed no significant association with placebo treatment intensity in either treatment arm ($\rho = -0.03$, $P = 0.57$ in placebo-treated villages; $\rho = 0.3$, $P = 0.59$ in azithromycin-treated villages; Supplementary Figure 9C)."

4. Likewise—could this be done not with the gene abundance, but with actual numbers of children (or proportions) with resistance? Is it possible that a village with 50 reads/million, for example, only had one such child? Could 2 children with resistance in one village account for more reads than 5 in another?

Given the preceding concern, it's good the analyses employ nonparametrically-derived p-values.

Response: This is a good question. In response to reviewers' comments, we interrogated our procedures and learnt that while samples from later timepoints were pooled, we had access to samples from 10 children per village sequenced individually at this timepoint.

We found that village-level mean resistance was representative of the underlying distribution within villages rather than being driven by a few highly resistant children. The proportion of children with detectable resistance (non-zero reads) was strongly correlated with village mean resistance at both baseline (Pearson $R = 0.58$, $p = 0.022$ for placebo; $R = 0.51$, $p = 0.05$ for azithromycin) and 24 months ($R = 0.74$, $p = 0.0018$ for placebo; $R = 0.42$, $p = 0.12$ for azithromycin). This indicates that higher village-level resistance reflects broader distribution of resistance across children rather than extreme values in a small number of individuals. We have now added Supplementary Figure 11 showing individual-level resistance patterns and their relationship with village-level means.

Lines 543-545 (Methods):

"Community-level means were highly correlated with the proportion of children with detectable resistance within villages (Supplementary Figure 11)"

5. I didn't see anything about nasopharyngeal swabs—what about their results?

Response: We thank the reviewer for pointing this out. As the reviewer noted, nasopharyngeal swabs were collected to assess phenotypic macrolide resistance in *Streptococcus pneumoniae* isolates, in addition to the rectal swab metagenomic analysis that forms the primary focus of this manuscript.

We have added Supplementary Figure 8, which presents the relationship between geographic treatment intensity and the proportion of *S. pneumoniae* isolates resistant to erythromycin. Consistent with the genotypic resistance findings from rectal swabs, we observed no significant associations between geographic treatment intensity and phenotypic macrolide resistance in *S. pneumoniae* at 24 months. Notably, pneumococcus was only isolated from 13 (azithromycin) or 14 (placebo) of the 15 villages in each treatment arm, resulting in slightly smaller sample sizes for this analysis.

Lines 207-225 (Results):

"To complement the metagenomic analyses of resistance determinants in rectal swabs, we assessed phenotypic macrolide resistance in Streptococcus pneumoniae isolated from nasopharyngeal swabs collected at 24 months. We evaluated the relationship between geographic treatment intensity and the proportion of S. pneumoniae isolates resistant to erythromycin using Spearman rank-order correlation, with statistical significance assessed through permutation tests. Pneumococcus was isolated from 14 of 15 AMR-monitoring villages in the placebo treatment arm and from 13 of 15 in the azithromycin treatment arm, respectively. Consistent with the genotypic resistance findings, no significant associations were observed between geographic treatment intensity of azithromycin and phenotypic macrolide resistance in either placebo-treated villages ($\rho = -0.13$, $P = 1.0$) or azithromycin-treated villages ($\rho = -0.36$, $P = 0.36$; Supplementary Figure 8A). The negative control exposure analysis using placebo treatment intensity similarly showed no significant associations ($\rho = -0.16$, $P = 0.76$ in placebo-treated villages; $\rho = -0.17$, $P = 0.91$ in azithromycin-treated villages; Supplementary Figure 8B). These phenotypic resistance data in a clinically relevant respiratory pathogen corroborate the absence of detectable geographic spillover effects observed in the metagenomic analyses."

Lines 552-562 (Methods):

*"Nasopharyngeal swabs were collected from 10 randomly selected children per village at 24 months, stored in skim milk-tryptone-glucose-glycerin (STGG) medium, and shipped to the University of California, San Francisco for processing. Pneumococcus isolation and antimicrobial susceptibility testing were performed at ARUP, a CLIA-certified laboratory, using standard broth microdilution methods and Clinical and Laboratory Standards Institute breakpoints; further details are provided in the supplementary appendix of Doan et al. The proportion of erythromycin-resistant isolates was calculated for each village where pneumococcus was successfully isolated. Associations between geographic treatment intensity and the proportion of macrolide-resistant *S. pneumoniae* were evaluated using Spearman rank-order correlations, with statistical significance assessed using conditional permutation tests as described above."*

Reviewer #1 (Remarks on code availability):

There is certainly enough information here to reproduce the results.

Reviewer #2 (Remarks to the Author):

Review Comments

1. The study relied on data from metagenomic deep sequencing performed on pooled rectal swab samples from 40 children per village. We know pooled samples suffers from several limitations such as masking of individual-level variations in resistance profiles, decreased diagnostic power/sensitivity, possible discrepancies with metagenomic and phenotypic characterizations, etc. Could the authors justify the use of this approach.

Response: We agree with the reviewer that pooled samples have inherent limitations, including the masking of individual-level variation and potential sensitivity concerns. Samples in the trial were pooled at later timepoints for cost-effectiveness in characterizing community-level resistance patterns across 30 villages. However, at baseline and 24 months, we had access to samples from 10 children per village that were sequenced individually. We have now revised our methods to clarify this distinction.

Lines 520-522 (Methods):

"Rectal swabs were collected from a random sample of 10 children aged 1 to 59 months in each AMR monitoring village at baseline and at 24 months. DNA was extracted from *each* sample, and its concentration was quantified and normalized for sequencing library preparation."

2. AMR was quantified as village-level normalized read counts of macrolide–lincosamide–streptogramin (MLS) resistance determinants at AMR-monitoring villages. How can one distinguish between active and inactive genes? As we may know, normalized read counts only quantify the presence of resistance genes, not whether they are actively transcribed and producing a functional protein.

Response: This is a good point. While, as noted by the reviewer, normalized read counts from metagenomic sequencing do not directly measure whether these genes are actively transcribed or producing functional proteins, our analysis focused on genotypic resistance (gene presence) as the primary outcome because it represents the genetic potential for resistance within the microbial community. However, we acknowledge that this approach does not distinguish between transcriptionally active and inactive genes.

To provide functional validation of our genotypic findings, we analyzed phenotypic macrolide resistance in *Streptococcus pneumoniae* isolates from nasopharyngeal swabs, which we have now included as Supplementary Figure 8 (see also our response to Reviewer 1, Comment 5). While phenotypic resistance is further downstream than genotypic resistance, we acknowledge that treatment failure, rather than phenotypic resistance itself, represents the clinically relevant endpoint, as noted in the discussion.

Lines 389-391 (Discussion):

"Although we examined both genotypic and phenotypic resistance, these measures serve as indicators of resistance potential rather than direct evidence of clinical treatment failure."

Conclusion/ Recommendation

The manuscript addresses the important problem of AMR. It investigated geographic spillover of AMR to untreated populations in mass drug administration (MDA) of azithromycin. It assessed between-village geographic spillover effects of genotypic resistance to macrolides and other antibiotic classes in rectal swabs collected from 1200 children in 30 monitoring villages in Niger after two years of MDA in 594 surrounding villages. The key finding suggested that azithromycin MDA-induced selection of macrolide AMR is localized to treated villages without extending to children in neighboring, untreated villages, thus mitigating previous concerns about geographic spillover of AMR to untreated populations in MDAs. Thus, this manuscript makes an important contribution to our knowledge on the transmission dynamics of AMR in particular and generally, the public health implications and potential risks of azithromycin MDA to children.

The study is clearly described, and the results are accurately presented. It should therefore be considered for publication, subject to consideration/clarification on the above 2 comments

Response: We are grateful for the reviewer's thoughtful comments, which have helped us substantially strengthen the manuscript.

Reviewer #3 (Remarks to the Author):

This is an important secondary analysis of the MORDOR mortality trial examining the potential geographic AMR spillover of MDA azithromycin for reducing childhood mortality. The methodology carefully lays out various statistical approaches to overcome limitations inherent in the original study design and data collection. The findings focus on an additional, small random sample of 30 villages, with 15 villages receiving MDA of azithromycin and 15 villages receiving placebo, amidst the 594 surrounding villages where the mortality benefit of MDA azithromycin was assessed. This monitoring occurred only in Niger, where the most pronounced benefits of lowering childhood mortality were found, in the original MORDOR study that also included study sites in Malawi and Tanzania.

The biological plausibility of intra-community-level transmission of AMR determinants of resistance to azithromycin is not in question, as this clearly has been documented in another study of MORDOR data. It is helpful to have documented that inter-community spread of AMR may not have been a concern in the conduct of this MDA azithromycin trial, where the treated population were children aged 1-59 months. The relevant take-away questions, of course, are why. Is it because the factors that spread macrolide resistance are not present between villages or the sample size for AMR monitoring was too small to detect a difference?

Response: Thank you for this thoughtful summary and synthesis of the analysis. We address each comment in detail in our point-by-point responses below. The updates have improved the manuscript.

1. It would be clearer to use "genotypic resistance" related to this study's findings, to avoid confusion, as this study did not address phenotypic resistance. As you know, in some cases, phenotypic resistance to macrolides may be observed when genotypic resistance is not detected by standard genetic tests.

Response: After consistent comments from all reviewers on this topic, we expanded our analysis to include phenotypic resistance results (Supplementary Figure 8), so we've kept the title and explicitly distinguished genotypic and phenotypic resistance in the text for clarity.

2. **Introduction** (page 2): The discussion of prior MORDOR data tracking macrolide-resistant determinants in azithromycin-treated villages, both among children in the study vs. others, would offer useful context. It would also better anticipate the much-needed section in the Discussion as to why macrolide-resistant determinants were not detected to have spread across villages in this study. Policymakers should not come away from this study imagining that somehow it is not biologically plausible that AMR spreads via humans, food products, or the environment.

Response: We thank the reviewer for this comment and agree that providing context about prior MORDOR AMR findings and the biological plausibility of resistance spread would strengthen the introduction and better frame our findings for policymakers. In response to this suggestion, we have expanded the introduction to include a discussion of prior evidence on macrolide resistance from MORDOR and the mechanisms by which AMR can spread between populations.

Lines 82-87 (Introduction):

"*Modeling studies suggest that antimicrobial resistance can spread between populations through person-to-person transmission of resistant bacteria, with even modest interactions between populations potentially facilitating spread. Concerns have been raised about the spillover of AMR from treated to untreated groups within communities receiving azithromycin MDA, with some evidence suggesting resistance may persist and potentially spread to untreated populations.*"

3. Measure of azithromycin treatment intensity (page 4): "We quantified geographic treatment intensity at any point in the study region as the cumulative number of doses of azithromycin distributed in mortality-monitoring villages over 24 months (Figure 1C), weighted by the inverse of the distance to that point. We used an inverse distance weighting approach with a linear decay

function to balance the influence of nearby and distant treated villages, aligning with the expected antimicrobial resistance spread.”

How does the “inverse distance weighting approach” account for factors that a more detailed investigation of transportation routes, mobility patterns, social exchange networks, administrative boundaries, or geographical barriers (waterways or mountainous terrain) might have? As the authors note in the Introduction, population mobility can be a fundamental driver of spread.

Response: We agree that our inverse distance weighting approach is a simplified representation of connectivity that does not explicitly capture mechanisms that may influence antimicrobial resistance spread. We have expanded the discussion about this limitation and also note that in the absence of such detailed data for rural Niger, we employed gravity models as a sensitivity analysis.

Lines 388-389 (Discussion):

“The distance decay is also assumed to be uniform across the study area, without accounting for potential variations in ease of connectivity and travel time between villages or geographic factors such as terrain and remoteness. This assumption may oversimplify some spatial relationships, *and the incorporation of high-resolution data on mobility patterns and social networks could strengthen the sensitivity.*”

Lines 490-492 (Methods):

“Gravity models are widely used to describe human movement across spatial scales, *particularly in settings where high-resolution data on mobility patterns, transportation routes, or social networks may be limited.* They have recently been applied to model population mixing and mobility in infectious disease transmission dynamics.”

4. What overlap was there with MDA for trachoma in the Niger study region, and might that have influenced the measure of azithromycin treatment intensity? Is MDA for trachoma a potential confounder to observations made in this secondary analysis?

Response: According to the GET2020 database, which monitors trachoma, the last round of MDA for trachoma in the Dosso region occurred in 2009, approximately 5 years before the MORDOR trial baseline measurements in December 2014. This provides a substantial washout period between any historical trachoma MDA and our study period. Importantly, the randomized design of the trial allows us to take a closer look at confounding, as discussed further in our response to comments 10 and 14, below.

Methods: Several additional methodological details would help the reader evaluate the study:

5. With only 15 placebo villages, what difference was the study powered to detect? Although demonstration of evidence of genotypic resistance in treated communities is helpful context, the focus here should be on the placebo communities.

Response: We agree that power considerations are crucial for interpreting our findings, particularly in placebo villages where spillover effects would be most apparent. Unfortunately, we are unaware of established methods for power calculations in studies of geographic spillover effects with spatial clustering and randomization. Developing appropriate power calculations would require complex simulations that appropriately account for the spatial structure of settlements, correlation structures between villages, and the decay of spillover effects with distance—perhaps an area for future methodological research.

We acknowledge that 15 placebo monitoring villages may limit our ability to detect spillover effects if they exist. To help provide context for the precision of our findings, we conducted a supplementary analysis that fit a log-linear model (Lines 263-277; Results) to estimate the fold-change in resistance that corresponds to increased azithromycin exposure in nearby villages. We have now expanded the Discussion to acknowledge this limitation more explicitly.

Lines 354-361 (Discussion):

"The upper bound of our post-hoc confidence intervals (up to 1.9-fold increase) suggests we cannot definitively rule out modest spillover effects. Appropriate power calculations for geographic spillover with spatial clustering remain an open methodological challenge, requiring complex simulations to account for settlement patterns, between-village correlations, and distance-decay functions. Future studies with larger monitoring samples and methodological advances in power estimation for spatially-structured spillover effects would strengthen inference about small-magnitude geographic spillover effects."

6. The authors should provide more details about the MLS genotypic testing. Does this test for all known genotypic mutations leading to macrolide resistance? What level of detection is considered significant? On page 10, gene fraction >80% is noted – is this standard?

Response: We thank the reviewer for this important suggestion. In response, we expanded the methods section and added references to the detailed laboratory methods published previously (doi.org/10.1056/NEJMc1901535; doi.org/10.1038/s41591-019-0533-0).

The 80% gene fraction threshold is the standard parameter used in the MEGARes/AMR++ pipeline to minimize false-positive classification of antimicrobial resistance genes. The primary outcome was the normalized abundance of resistance genes (reads per million), analyzed as a continuous variable rather than using a binary significance threshold. The metagenomic approach identifies a comprehensive range of MLS (macrolide-lincosamide-streptogramin) resistance determinants present in the MEGARes database at the time of analysis, including macrolide resistance efflux pumps, phosphotransferases, and 23S rRNA methyltransferases.

Lines 536-540 (Methods):

"A sample was determined to be macrolide resistant if at least one ARD in the MLS class was detected. Microbiome sequencing reads can be accessed at <https://microbiomedb.org/mbio/app>. Detailed laboratory protocols for sample processing, DNA extraction, library preparation, and sequencing are provided in detail in previously published studies."

7. **Results:** Random sample of 30 additional villages (page 3): It would be helpful to the reader to understand why AMR monitoring was not embedded as a subset, as opposed to an additional sampling, of the villages.

Response: We thank the reviewer for this suggestion that clarifies the study design for the reader. We have revised the discussion in response.

Lines 333-335 (Discussion):

"A random sample of 30 villages were separately randomized to avoid potential effects of intensive sample collection on the primary mortality outcome. Cluster randomization over 594 villages created variation in geographic treatment intensity across the independently sampled 30 AMR monitoring villages, and permuting the random assignments using a conditional permutation test allows for an exact, non-parametric test for the presence of spillover effects between villages"

8. Figure 1B suggests that the separate random sample of 30 villages are heavily concentrated in one part of the Dosso region of Niger.

Should the reader know whether this geographical concentration correspond to any factors (such as the aforementioned concerns with distance alone) that might have influenced the true dynamics of resistance spillover?

Response: The 30 AMR monitoring villages were selected as a simple random sample from the original sampling frame of eligible villages. The apparent clustering in Figure 1B likely reflects the

underlying spatial distribution of settlements in the Dosso region. We have added clarifying text to address this in the Discussion.

Lines 348-352 (Discussion):

"Although simple random sampling resulted in some geographic clustering that reflects underlying settlement patterns, the 30 monitoring villages spanned the full range of observed treatment intensity and showed similar distributions of geographic factors such as distance to health posts (Supplementary Figure 10) compared to the 594 mortality-monitoring villages"

9. Inverse distance weighting performs less well when sample points cluster. Is this a methodological concern here (or not) given the apparent geographical clustering of the randomly selected villages in the AMR monitoring sample within the Dosso region?

Response: This is an interesting methodological question. While inverse distance weighting can perform poorly when sample points cluster during spatial interpolation to unsampled locations because clustered samples provide redundant information while leaving sparse areas poorly represented. However, we are not interpolating values but rather calculating a treatment intensity metric at each point based on the spatial distribution of the 594 mortality monitoring villages relative to the point.

The apparent geographic clustering of AMR monitoring villages likely reflects the underlying settlement patterns in the Dosso region. Since we used simple random sampling from all eligible villages in the study area, this clustering is expected and ensures our sample represents areas with varying settlement densities. Since the treatment intensity was estimated independently for each AMR-monitoring village based on its unique spatial position relative to the 594 mortality-treatment villages, each AMR-monitoring village has its own distinct exposure profile, regardless of how close it is to other AMR-monitoring villages. We believe this fundamental difference in application, quantifying exposure at known locations rather than predicting values at unmeasured locations, means the methodological concerns about IDW performance with clustered samples should not bias this analysis.

10. Results—Spillover effects of azithromycin MDA to nearby villages (page 4): Could the authors address potential confounders that likely will arise in the mind of readers?

Does the fact that MLS resistance determinants remained low and at baseline levels in placebo-treated AMR monitoring villages at 24 months provide sufficient assurances (or just a higher baseline) from confounding factors, such as the frequent administration of antibiotics for childhood ailments in these settings?

Response: The stability of MLS resistance in placebo villages—remaining low at both baseline and 24 months—demonstrates minimal influence of time-varying factors such as differential changes in healthcare-seeking behavior or antibiotic access during the study period.

However, our key test for confounding is the negative control exposure analysis, which was specifically designed to detect spurious associations arising from confounding or other non-causal sources of variation in AMR unrelated to azithromycin MDA. We have expanded the description of this approach in the Methods section to emphasize its role in detecting confounding.

Lines 603-609 (Methods):

"Randomization of treatment assignments in the mortality monitoring villages balanced all potential confounders—both measured and unmeasured, including spatially structured factors—across treatment arms. The geographic treatment intensity of placebo is identically structured to the geographic treatment intensity of azithromycin, yet has no causal effect on AMR, making it an ideal negative control exposure that captures all sources of confounding that affect both treatment intensity measures similarly."

Regarding background macrolide exposure specifically, based on health post dispensing data from this region (2019-2023, unpublished), macrolides accounted for less than 1% of antibiotics dispensed for childhood ailments, with beta-lactams predominating at approximately 60%. This low background rate of macrolide use makes it unlikely to substantially confound our spillover assessment.

11. Macrolide antibiotics are one of the more commonly used families of antibiotics in food animal production in Sub-Saharan Africa (<https://pmc.ncbi.nlm.nih.gov/articles/PMC10830768/>). Could this have potentially influenced observations made in the study? More intensive animal food production systems may use antibiotics to a greater degree. Would these also be randomly distributed throughout the Dosso region?

Response: We thank the reviewer for raising this consideration about potential antimicrobial use in animal production. Systematic data on animal antibiotic use was not collected in the trial, and to our knowledge, there are no published studies characterizing veterinary antibiotic use specifically in this part of Niger. Anecdotally, animal husbandry in the study region is primarily subsistence-level, with domestic chickens being the predominant livestock, rather than the more intensive production systems where antibiotic use tends to be higher. While colistin is the primary antibiotic used for poultry in this setting, we cannot rule out some macrolide use in animal production.

However, regarding the impact of spatially variable animal antibiotic use on the results, we refer the reviewer to our response to comment 10, where we describe how the negative control exposure analysis and cluster randomization design provide safeguards against unmeasured spatially-structured confounders of this nature.

12. In this section, it is also noted that: "A modest correlation between geographic treatment intensity and MLS resistance in azithromycin-treated villages was observed at baseline ($p=0.30$) and remained similar at 24 months ($p=0.32$), though neither was significant under a conditional permutation test ($P=0.21$ at baseline, $p=0.95$ at 24 months)." Actually, none of the p-values are significant at the $P<0.05$ level. How do we know that the sensitivity of this measure to detect AMR spillover is sufficient for doing so?

Response: Please see our response to comment 5, where we have addressed power considerations.

13. "Metagenomic deep sequencing was performed on pooled rectal swab samples from 40 children per village..." How might have the pooling of rectal swab samples affected the validity that might have been resulted from an analysis of the individual samples? Were these samples transported to local laboratories or to a more distant central laboratory for processing? Were there any systematic biases in metagenomic sequencing that might be introduced from transport and processing of these samples that required correction (<https://pmc.ncbi.nlm.nih.gov/articles/PMC6739870/>)?

Response: This is a good question. Following reviewers' comments, we clarified that samples at 24 months were sequenced individually for 10 children per village rather than pooled as in later study visits (apologies for the error in the last draft). We added Supplementary Figure 11, which demonstrates that village-level means from individual samples at 24 months provide valid representations of resistance patterns, and edited the methods accordingly. Regarding transport, all samples were stored in preserving media and immediately placed on ice in the field and transported immediately upon collection for transport to the University of California, San Francisco. We agree that the reference cited by the reviewer highlights important considerations for metagenomic studies. However, all samples for this study were processed in the same laboratory using identical protocols. Village treatment assignments were randomized, and laboratory personnel were masked to treatment allocation. Samples from both treatment arms and all time points were processed in parallel. These measures ensured that any technical variation would be distributed equally across treatment groups and would not introduce systematic bias that would require correction. The detailed bioinformatics pipeline, including quality control steps, is provided in the supplementary appendix of Doan et al. (2019) and has been referenced in the current manuscript (lines 539-541).

14. Results—Negative control exposure analysis (page 4): “Using the same methods used to estimate azithromycin geographic treatment intensity, but with the cumulative number of placebo doses distributed instead of the cumulative azithromycin doses, the negative control exposure analysis was designed to detect spurious associations arising from unmeasured confounding, or other non-causal sources of variation in AMR that are unrelated to azithromycin MDA [22,23].” Does not negative control exposure analysis only adjust for some types of unmeasured confounding, but not other unmeasured confounding (e.g., macrolides used in livestock raising in some villages but not others)?

Response: This question helps clarify the scope of the negative control exposure analysis. The reviewer is correct that negative control exposures do not universally adjust for all types of unmeasured confounding. However, in our specific study design, the negative control exposure has a broader protective scope than in typical observational settings.

A key feature of our study is that randomization of treatment assignments in the mortality monitoring villages broke any causal pathways between potential confounders and the treatment intensity measures. As a result, any factors that might confound the relationship between geographic treatment intensity and AMR would affect both the azithromycin and placebo treatment intensity measures in the same way. This makes the geographic treatment intensity of placebo a perfect negative control exposure in our context: it detects all measured and unmeasured confounding, including spatially structured confounding, because the treatment intensity of placebo mimics the spatial structure of the treatment intensity of azithromycin due to the balanced randomization. To address the specific example raised by the reviewer regarding macrolide use in livestock: if such village-level factors were associated with AMR, they would be captured by our negative control because the randomization ensures that villages with higher livestock macrolide use would be equally represented in both treatment arms. We have now clarified this salient point in the methods section.

Lines 603-609 (Methods):

“Randomization of treatment assignments in the mortality monitoring villages balanced all potential confounders—both measured and unmeasured, including spatially structured factors—across treatment arms. The geographic treatment intensity of placebo is identically structured to the geographic treatment intensity of azithromycin, yet has no causal effect on AMR, making it an ideal negative control exposure that captures all sources of confounding that affect both treatment intensity measures similarly.”

15. Results—Non-parametric analysis of geographic spillovers (page 5): “In the presence of AMR spillover, we hypothesized that there would be a stronger correlation between azithromycin doses delivered and MLS resistance at smaller distances since closer proximity increases the likelihood of transmission of resistant strains between villages. The correlations between MLS resistance determinants and geographic treatment intensity within most distance bands were low.” The discrete distance bands were within 10 km, 10-20 km, and 20-30 km of each AMR monitoring community over 24 months. Does this lack of a gradient suggest the insensitivity of the inverse distance weighting approach in capturing this?

Response: This is an interesting point in considering whether the lack of a distance gradient might indicate insensitivity of our inverse distance weighting (IDW) approach. We believe that the IDW and distance band approaches are methodologically complementary as they operationalize spatial proximity in fundamentally different ways, yet both independently converge on the same conclusion. If spillover were present but our IDW approach was simply too insensitive to detect it, we would expect the more granular distance band analysis to reveal the effect, particularly in the 0-10 km band where transmission would be strongest. That neither approach detects an effect suggests the null result reflects the true absence of spillover rather than measurement insensitivity.

16. Results—Sensitivity analyses (page 5): Relying on physical distance, inverse distance weighting might not adequately capture actual functional connectivity. How might network-based models or the gravity model discussed offset these limitations? The “gravity models...were adapted to assess macrolide resistance spread by using an inverse distance-squared weighted sum of azithromycin doses in mortality monitoring villages as an alternative geographic treatment intensity measure.” How is this gravity model different from the inverse distance weighting?

Response: The reviewer raises an important point about capturing functional connectivity. We agree that network-based models incorporating empirical data on social networks, mobility patterns, or transportation routes could better represent actual connectivity between villages. However, the trial did not measure metrics to assess the social networks or connectivity of participating villages. In the absence of such data, gravity models have successfully been used in the literature to model population mixing and mobility in infectious disease transmission dynamics. Our gravity model used the inverse distance-*squared* to weight the number of neighboring doses, as opposed to a simple inverse-distance weighting, as discussed in the methods (lines 489-493).

17. Rather than examine AMR spillover in proportion to the absolute number of azithromycin doses distributed, might an alternative approach have considered the proportion of the population treated, or even better, the proportion of the treated population that might transit between villages?

Response: We thank the reviewer for raising this important methodological point. The reviewer is correct that proportion-based measures could provide additional insights into spillover dynamics. Our choice of an absolute dose-based distance measure was driven by two key constraints in the study design.

- First, the trial restricted enrollment to villages with populations between 200-2000 residents, explicitly excluding larger villages and urban areas. This restriction makes population-based proportions unreliable as denominators, as we lack consistent population estimates across the region. This limitation is further compounded by edge effects: we do not have population estimates for areas outside the study boundaries, which would be necessary to properly calculate treatment proportions in the broader geographic context.
- Second, while we agree that incorporating data on between-village transit patterns would be highly valuable for understanding resistance transmission dynamics, such mobility data was not collected in this trial. Human movement patterns could indeed modify spillover effects, and we appreciate the reviewer highlighting this as an important consideration for future study designs.

Lines 361-384 (Discussion):

“Another potential limitation is the choice of spillover measure: we assume the potential for AMR spillover is proportional to the absolute number of azithromycin doses distributed rather than the proportion of the population treated. *This approach was chosen because the trial’s restriction to non-urban villages of 200-2000 residents, with exclusion of larger settlements, made population-based proportions unreliable due to incomplete population sampling and edge effects.*”

18. Given that those treated with MDA azithromycin were aged 1-59 months, might this segment of the population not be particularly mobile between villages at that age? Might that explain the finding of no AMR spillover between villages? If that is the case, might it be important to note that the findings in this secondary analysis of MORDOR may not be generalizable to, say, MDA trachoma in the general population?

Response: The reviewer raises two important considerations about the mobility patterns of young children and the generalizability of our findings that we have now addressed in the Discussion.

Lines 391-403 (Discussion):

“...our findings may not be directly generalizable to trachoma control programs, which differ in two key aspects: trachoma MDA typically targets entire communities rather than only children aged 1-

59 months, and treatment frequency is generally annual rather than biannual. While children aged 1-59 months may not travel independently between villages, villages in this setting regularly share community infrastructure including schools, markets, and health posts, providing opportunities for contact and potential transmission of resistant bacteria between children from different villages. The lack of observed spillover despite these opportunities may reflect insufficient contact intensity or duration for sustained between-village transmission, limited persistence of resistant strains without ongoing selective pressure, or rapid strain turnover in young children's gut microbiomes. The broader population coverage and higher mobility of adults in trachoma programs could potentially alter spillover dynamics through increased treatment intensity and expanded networks of transmission."

19. Results – Figure 3C. The authors dismiss a statistically significant finding at the 20-30 km radius for the placebo groups because it was not consistent with their hypothesis that there would be strong correlations at closer distances. Might this finding not simply undermine the hypothesis, warranting more careful interpretation of the findings and subsequent Discussion?

Response: We thank the reviewer for this comment on this significant finding warranting more careful interpretation. Following this suggestion, we have expanded our discussion of this result to provide a more nuanced interpretation that considers both the statistical significance and the broader pattern of findings across all distance bands.

Lines 243-249 (Results):

"Although there was a significant correlation in placebo-treated villages in the 20-30 km band, this finding should be interpreted cautiously, given that correlations at closer distances were weaker or negative ($p = -0.07$ at 0-10 km, $p = 0.24$ at 10-20 km), inconsistent with a spillover mechanism where proximal villages would show stronger effects. Moreover, this represents the only significant finding among 12 correlation tests across distance bands, timepoints, and treatment arms, compatible with a chance finding."

20. **Discussion** (pages 6-8): Some of the foregoing issues may warrant further explanation in the Discussion section, including:
- limited sample size of AMR monitored villages;
 - reasons for lack of finding AMR spillover between villages in this context, including azithromycin-treated group being 1-59 months of age;
 - generalizability (or not) given MDA trachoma involves the entire population, notably age segments of the population more mobile across villages.

Response: We thank the reviewer for these helpful suggestions. We agree that these points warrant expanded discussion, and we believe addressing them has strengthened the manuscript substantially. We have revised the Discussion section to address each of these important considerations:

a. Limited sample size: Lines 348-354

b & c. Age range and generalizability to trachoma programs: Lines 391-403

21. The Discussion notes that the "findings suggest that current AMR monitoring strategies in sentinel villages...are likely sufficient." Whether this follows from the findings may depend on responses above – was the study sufficiently powered? Would more targeted sentinel communities, based on proximity and human mobility, actually be a better method compared to random sampling?

Response: We agree that this conclusion about monitoring strategies should be qualified by the study's power limitations and design considerations. In response to this comment, we revised this statement to be more appropriately cautious and power considerations have been discussed elsewhere in the discussion (lines 352-354)

Lines 315-322 (Discussion):

"Our findings support current AMR monitoring strategies in sentinel villages, as we found no evidence of geographic spillover, which could cause these surveillance strategies to underestimate AMR selection during the initial years of MDA implementation."

Reviewer #4 (Remarks to the Author):

Response: Thank you for your contributions to a very thorough, constructive review.

REVIEWERS' COMMENTS

Reviewer #1 (Remarks to the Author):

Great to see the additional by-village change analyses.

Response: We thank the reviewer for suggesting this nice complementary analysis and their constructive feedback throughout.

Reviewer #3 (Remarks to the Author):

We appreciate the authors' efforts to address the methodological concerns raised about this paper. Most were addressed with useful caveats and acknowledged limitations in the text. The absence of spillover is not biologically plausible, in general, but only possible in specific circumstances. The question is what this study sheds insight regarding what are those circumstances and whether current AMR monitoring strategies are sufficient, compared to better or more comprehensive approaches to monitoring. The difficulty in assessing whether the study is sufficiently powered, the geographical clustering, the lack of data on actual ties between communities (mobility of people), and the lack of information about potential confounders like the use of antibiotics in local livestock raising (which unfortunately in West Africa is prevalent among both small farmers and intensive farming operations — <https://onlinelibrary.wiley.com/doi/full/10.1002/agr.21770>) make determining what those insights are challenging. Taking into account the study's limitations, it might be useful to rethink whether one should be calling for continuing just more of the same AMR monitoring strategies or improved vigilance ("Our findings support current AMR monitoring strategies in sentinel villages, as we found no evidence of geographic spillover, which could cause these surveillance strategies to underestimate AMR selection during the initial years of MDA implementation..."). Given what is at stake regarding AMR and child mortality, the absence of evidence may not be evidence of absence, and the paper's findings should be framed and interpreted accordingly.

Response: We thank the reviewer for this thoughtful comment about the interpretation and framing of our findings. The reviewer is correct that the absence of evidence is not evidence of absence, particularly given the critical stakes around both antimicrobial resistance and child mortality. In response to this comment, we have revised our interpretation throughout the manuscript to be more nuanced and measured.

Lines 362-371 (Discussion):

"Spillover effects at smaller spatial scales, such as within-household or within-village, could be present [20,29], but substantial, unmeasured indirect effects of mass drug administration on resistance are unlikely beyond the effects observed within treated communities over a 24-month period (four MDA treatments). Our findings suggest that current sentinel village AMR monitoring strategies, which do not explicitly account for between-village spillover effects, likely captured the AMR selection attributable to MDA during the initial years of program implementation in this setting."

Lines 494-500 (Discussion):

"Future analyses of large-scale, azithromycin MDA trials in the region should *enable tests for spillover effects* over longer periods and *at smaller spatial scales*. In summary, there was no evidence of between-village spillover of macrolide resistance following azithromycin MDA, suggesting that AMR effects are largely localized to treated villages without *substantial* spreading to neighboring areas *at the spatial and temporal scales evaluated*."

Lines 36-38 (Abstract):

"Azithromycin MDA-induced macrolide resistance *appears* localized to treated villages without extending to children in neighboring, untreated villages, mitigating some concerns about geographic spillover of AMR to *nearby* untreated villages at 24 months."

Reviewer #4 (Remarks to the Author):

Response: Thank you for your contributions.